# Entrectinib Induces Apoptosis and Inhibits the Epithelial–Mesenchymal Transition in Gastric Cancer with NTRK Overexpression

**DOI:** 10.3390/ijms23010395

**Published:** 2021-12-30

**Authors:** Sung-Hwa Sohn, Hee Jung Sul, Bum Jun Kim, Hyeong Su Kim, Dae Young Zang

**Affiliations:** 1Hallym Translational Research Institute, Hallym University Sacred Heart Hospital, Anyang 14066, Korea; iisupy@korea.ac.kr (S.-H.S.); glwjd82@naver.com (H.J.S.); 2Department of Internal Medicine, Hallym University Medical Center, Hallym University College of Medicine, Anyang 14068, Korea; getwisdom1025@gmail.com (B.J.K.); nep2n74@gmail.com (H.S.K.)

**Keywords:** NTRK, entrectinib, β-catenin, ECAD, CD44, NFκB

## Abstract

Tropomyosin receptor kinase (TRK) and receptor tyrosine kinase (RTK class VII) expression are important in many human diseases, especially cancers, including colorectal, lung, and gastric cancer. Using RNA sequencing analysis, we evaluated the mRNA expression and mutation profiles of gastric cancer patients with neurotropic tropomyosin receptor kinase (NTRK) 1-3 overexpression (defined as a ≥2.0-fold change). Furthermore, we screened eight TRK inhibitors in NCI-N87, SNU16, MKN28, MKN7, and AGS cells. Among these inhibitors, entrectinib showed the highest inhibitory activity; therefore, this drug was selected for analysis of its therapeutic mechanisms in gastric cancer. Entrectinib treatment induced apoptosis in NTRK1-3-expressing and VEGFR2-expressing NCI-N87 and AGS cells, but it had no effect on NTRK1-3-, VEGFR2-, TGFBR1-, and CD274-expressing MKN7 cells. SNU16 and MKN28 cells with low NTRK1-3 expression were not affected by entrectinib. Therefore, a mechanistic study was conducted in NCI-N87 (high expression of NTRK1-3 but mutation of NTRK3), AGS (high expression of NTRK1-3) and MKN28 (low expression of NTRK1-3) gastric cancer cell lines. Entrectinib treatment significantly reduced expression levels of phosphorylated NFκB, AKT, ERK, and β-catenin in NCI-N87 and AGS cells, whereas it upregulated the expression levels of ECAD in NCI-N87 cells. Together, these results suggest that entrectinib has anti-cancer activity not only in GC cells overexpressing pan NTRK but also in VEGFR2 GC cells via the inhibition of the pan NTRK and VEGFR signaling pathways.

## 1. Introduction

Gastric cancer (GC) is the most prevalent malignant tumor in Korea. Metastasis is a major obstacle for improving long-term survival after surgical resection [1]. Thus, there is a need to identify effective therapeutic targets for metastatic GC. Neurotropic tropomyosin receptor kinases (NTRK1, NTRK2, and NTRK3) encode tyrosine receptor kinases (TRKA, TRKB, and TRKC), which induce many types of cancer pathogenesis through the activation of downstream signaling [2]. NTRK gene fusions are known to be oncogenic, while NTRK gene amplifications are associated with metastatic cancer [3]. NTRK gene amplification was also shown to result in TRK overexpression [3]. For the treatment of NTRK gene fusion, several TRK inhibitors have been developed. However, NTRK gene rearrangement is rare in cancer, so target therapy is rarely applicable.

The receptor tyrosine kinase (RTK) and Wnt/β-catenin pathway synergistically regulate essential developmental processes such as those of the vital organs (brain, heart, lung etc.) and skeleton [4]. RTK activates the Wnt/β-catenin pathway by inhibiting phosphoinositide 3-kinase (PI3K)/Akt-mediated glycogen synthase kinase 3 (GSK3) [5]. In addition, the Wnt/β-catenin pathway is known to drive the epithelial–mesenchymal transition (EMT) and cancer metastasis [6]. Cross-talk among the β-catenin, ERK/MAPK, and PI3K/Akt signaling pathways has been detected in many cancers, with simultaneous inhibition of apoptosis and blocking of differentiation [7,8]. Among these pathways, β-catenin is upregulated in cancer patients and was found to be associated with poor clinical prognosis [9]. Additionally, EMT is a molecular signature in neoplasia, which is strongly implicated in tumor cell invasion and metastasis [10,11]. In addition, EMT was characterized by the downregulation of epithelial markers such as E-cadherin (ECAD). Low or no expression of ECAD was reported in various types of epithelial cancer [10]. CD44 was involved in the EMT of cancer cells [12]. Overexpressed CD44 inhibit ECAD expression [13]. In addition, CD44 mediates cell–cell interactions and contributes to a worse prognosis for cancer patients [10,14]. CD44 promotes the proliferation, adhesion, and invasion of cancer cells, while overexpression of ECAD decreases the invasion of cancer cells [15]. 

NTRK mutation or amplification, such as non-fusion NTRK alteration, has been associated with a lack of response to NTRK inhibitors. Here, we performed an RNA sequencing analysis in 16 GC patients with amplification of NTRK1-3. We also screened eight TRK inhibitors in NCI-N87 (high expression of NTRK1-3 and VEGFR2, but mutation of NTRK3), AGS (high expression of NTRK1-3 and VEGFR2), MKN7 (high expression of NTRK1-3, TGFBR1, and CD274 but VEGFC mutation), SNU16 (low expression of NTRK1-3), and MKN28 (low expression of NTRK1-3) GC cell lines. Among the TRK inhibitors, entrectinib showed the highest inhibitory activity in GC cells with or without NTRK1-3 expression. We also evaluated the effects of entrectinib on apoptosis and EMT in a GC cell line, as well as its potential therapeutic mechanism.

## 2. Results

### 2.1. RNA Sequencing Analyses of 16 GC Patients with NTRK1-3 Overexpression

GC and adjacent normal tissues were obtained from 34 patients with total or subtotal gastrectomy who underwent initial surgery at Hallym University Sacred Heart Hospital (2015-I078) from March 2014 to July 2015. We performed RNA sequencing analysis of 34 GC patients (mostly stage III–IV). Then, we selected 16 GC patients overexpressing NTRK1–3, defined as a ≥2.0-fold change (Table 1 and Figure 1) [16]; the incidence was 47%, which was much higher than the rate reported in previous studies on 17–33 different tumor types (2.2–14.2%) [3,17]. Unlike other studies, the tumor tissue and adjacent normal tissue of 34 Korean GC patients aged 69 years (range: 44–87 years), mostly with stage II–IV disease, were analyzed. However, the small number of patients limited the interpretability of the results. These patients did not exhibit fusion or mutations of NTRK1-3. A Venn diagram was used to identify upregulated NTRK1-3 genes in 16 GC patients. NTRK1 was upregulated in nine patients, NTRK2 in nine patients, and NTRK3 in seven patients (Figure 1b). Three patients exhibited upregulation of all three NTRKs. In 16 GC patients with NTRK amplification, the expression levels of NOS2, CD274 (also known as B7-H1 and PD-L1), KDR (VEGFR2), CD44, VEGFA–C, CTNNB1, and TGFBR1 were upregulated (≥2.0-fold change), while those of MUC6 and NQO1 were downregulated (≤−2.0-fold change) (Figure 1c).

### 2.2. Effects of NTRK Inhibitors on GC Cells

We observed the inhibitory effects of three different forms of dovitinib and five TRK inhibitors on NCI-N87, SNU16, MKN7, MKN28, and AGS cells. The NTRK1 (≥20-fold), NTRK2 (≥40-fold), NTRK3 (≥50-fold), and VEGFR2 (≥150-fold) genes were more highly expressed in NCI-N87, MKN7, and AGS cells, compared with the other GC cell types (Figure 2). However, p-TRKA protein was only expressed in NCI-N87 and MKN7 cells. SNU16 and MKN28 cells did show low expression (≤1-fold) of the NTRK1-3, VEGFR2, TGFBR1, or CD274 genes, compared with the other GC cell types. However, the pan-TRK and VEGFR2 proteins were more highly expressed in SNU16 cells compared with the MKN28 cell line. Notably, the TGFβR1, VEGFR2, and CD274 genes and proteins were highly expressed in MKN7 cells. In the five GC cell lines, entrectinib showed the best inhibition of cell viability regardless of gene and protein changes (Figure 3).

### 2.3. Effects of Entrectinib on Cell Apoptosis in GC Cells Harboring NTRK1/2/3, VEGFR2, TGFBR1, or CD274 Gene Amplification

To evaluate the effects of entrectinib on cell death in NCI-N87 (NTRK3, p.E211V; CCLE database), SNU16, MKN7 (VEGFC, p.G5G; CCLE database), MKN28, and AGS cells, we examined apoptosis by staining with annexin V-APC/PI, followed by flow cytometry (Figure 4) to assess early apoptotic and late apoptotic cell populations. Among all cell types, the apoptosis rates were highest for NCI-N87 and AGS cells with entrectinib treatment. The percentages of early and late apoptotic cells were 31.25% (vs. 10.68% in the control group) and 17.68% (vs. 4.09% in the control group) after exposure to entrectinib in NCI-N87 and AGS cells, respectively (Figure 4). Entrectinib induced more apoptosis in NCI-N87 and AGS with higher VEGFR2 expression, indicating that the cytotoxic effects of entrectinib might be due to activation of the VEGFR2 signaling pathway and not NTRK1-3. Apoptosis was seldom observed in SNU16, MKN7, and MKN28 cells (Figure 4). Although entrectinib was effective in cells expressing the NTRK mutation and the VEGFR2 gene, it did not induce apoptosis in cells with increased expression levels of the TGFBR1 and CD274 genes. CD274 overexpression is a statistically significant poor prognostic factor in multiple solid tumors (OS (HR = 1.58, 95% CI = 1.38–1.81, *p* < 0.000) and DFS/PFS (HR = 1.72, 95% CI = 1.26–2.33, *p* = 0.001)); overall survival analysis previously showed a statistically significant detrimental effect of PD-L1 in GC (HR = 1.56, 95% CI = 1.02–2.37, *p* = 0.040) [18].

### 2.4. Effects of Entrectinib on Gene and Protein Expression Levels in GC Cells

To further investigate the mechanism of action of entrectinib in NCI-N87 (high expression of NTRK1-3 but mutation of NTRK3), AGS (moderate expression of NTRK1-3), and MKN28 (low expression of NTRK1-3) cells, we analyzed the mRNA and protein expression levels in these cells (Figure 5 and Figure 6). qPCR revealed upregulation of EMT-inhibiting genes, such as ECAD and MUC6, in entrectinib-treated NCI-N87 (3.14 ± 0.05-fold and 4.33 ± 0.53-fold, respectively), AGS (1.97 ± 0.04-fold and 3.14 ± 0.22-fold, respectively), and MKN28 (1.84 ± 0.09-fold and 4.24 ± 0.33-fold, respectively) cells (Figure 5). Notably, the CD44 (0.58 ± 0.01-fold), CD44s (0.74 ± 0.12-fold), and CD44v9 (0.50 ± 0.02-fold) genes were only suppressed in entrectinib-treated AGS cells, while these genes were upregulated in entrectinib-treated MKN28 cells (4.14 ± 0.36, 1.43 ± 0.24 and 1.85 ± 0.18, respectively). However, changes in CD44, CD44s, and CD44v9 gene expression levels were seldom observed in entrectinib-treated NCI-N87 cells. Immunoblot analysis showed downregulation of phosphorylated NFκB (phospho-NFκB), phospho-ERK, phospho-AKT, VEGFR2, VEGF, Snail, and β-catenin in entrectinib-treated NCI-N87 and AGS cells; in contrast, NQO1 and GSK3 β protein expression was increased (Figure 6). ECAD protein was not expressed in AGS cells; thus, it was increased only in NCI-N87 and MKN28 cells. NTRK protein did not decrease despite treatment with entrectinib in NCI-N87 and MKN28 cells. However, p-TRKA protein expression was only decreased in NCI-N87 cells. p-TRKA protein was not expressed in AGS or MKN28 cells.

### 2.5. Effects of Entrectinib on Cell Migration in GC Cells

To further investigate the inhibitory effect of entrectinib on the migration ability of NCI-N87 (high expression of NTRK1-3 but mutation of NTRK3), AGS (moderate expression of NTRK1-3), and MKN28 (low expression of NTRK1-3) cells, we conducted a wound-healing assay of these cells. After 14 days, NCI-N87 control cells filled 55.78% of the wound area and entrectinib-treated NCI-N87 cells filled 5.18% of the wound. After 48 h, AGS and MKN28 control cells filled 100% and 100% of the wound area, respectively, and entrectinib-treated AGS and MKN28 cells filled 9.95% and 45.30% of the wound, respectively (Figure 7). These findings indicate that entrectinib decreases the mobility of GC cells. 

## 3. Discussion

NTRK1-3 amplifications are associated with metastatic cancer [3]. The major finding of the present study was that GC patients with NTRK1-3 amplification exhibited enhanced expression of NOS2 (iNOS), CD274 (PD-L1), KDR (VEGFR2), CD44, VEGFA–C, CTNNB1, and TGFβR1, while they exhibited reduced expression of MUC6 and NQO1 (to be included in these genes, the standard deviation of log2 expression differences had to exceed two, totaled over all samples). Therefore, after selecting an effective drug from the GC cell line with or without NTRK1-3 amplification, on the basis of cell viability, we examined the changes in genes and protein expression levels of these factors. We examined three forms of dovitinib and five TRK inhibitors from a TRK inhibitor screening library; among these compounds, entrectinib showed the best GC cell growth inhibition with or without amplification of NTRK1-3. Entrectinib induced apoptosis in NCI-N87 and AGS cells with high expression of NTRK1-3 and VEGFR2; however, entrectinib did not induce apoptosis in MKN7 cells with high expression of NTRK1-3, VEGFR2, TGFβR1, and CD274. Notably, TGFβ drives EMT, invasion, and migration [19,20]. The TGFβ signaling pathway was recently described as a potential mechanism of resistance for anti-CD274 checkpoint blockade [21,22]. CD274 upregulation also suppresses anti-tumor immunity in various human cancer types [23,24].

RTKs inhibit ECAD, which leads to the EMT but also activates Wnt/β-catenin signaling [25,26]. TRK and VEGF receptors belong to the RTK family. The TRKA signaling pathway is involved in cell growth and proliferation via ERK signaling [27,28]. TRKB activates the PI3K, PLCγ, and RAS-ERK pathway, resulting in cell differentiation and survival [26,27]. TRKC activate the PI3K/AKT pathway, preventing apoptosis and increasing cell survival [27,28]. Indeed, we observed that the NCI-N87 and AGS gastric cancer cells were more finely sensitive to the pan-TRK inhibitor entrectinib than MKN28. GC progression involves metastasis via Wnt/β-catenin signaling and EMT, which promotes invasion and regrowth in different organs. These pathways are biologically and clinically important for the development of anti-metastatic strategies [29,30]. ECAD bind β-catenin and influence its signaling and transcriptional activity in the nucleus [9]. ECAD loss, caused by the translocation of β-catenin into the nucleus, permits the regulation of transcriptional activity. The expression or nuclear localization of β-catenin indicates constitutive activation of the Wnt/β-catenin pathway [28]. The loss of ECAD expression is involved in EMT [31,32,33]. 

CD44 increases the metastatic potential of carcinoma cells [34], while reduced expression of ECAD is associated with differentiation and metastasis [35,36]. Cells with high expression of CD44 have drug resistance and high tumor-seeding ability through EMT [37], while ECAD genes are suppressed downstream to maintain EMT [29,38]. Cancer stem cell signatures enhance metastatic propensity [39] through cancer stem cell stemness via the PI3K/AKT, ERK/MAPK, and WNT/β-catenin pathways in colon, gastric, and prostate cancers [40,41]. Taken together, entrectinib is a multi-tyrosine kinase inhibitor that blocks VEGFR2 and NTRKs, as well as β-catenin, Snail, NFκB, AKT, and ERK. Therefore, entrectinib blocks angiogenesis, cancer progression, and EMT, while it induces apoptosis.

## 4. Materials and Methods

### 4.1. Cell Culture and Reagents

The human GC cell lines NCI-N87, SNU16, MKN7, MKN28, and AGS were obtained from the Korean Cell Line Bank (Seoul, Republic of Korea) and maintained in RPMI-1640 supplemented with 10% fetal bovine serum. The cells were cultured at 37 °C with 100% humidity and 5% CO_2_. LOXO-101, entrectinib, dovitinib, dovitinib lactate, dovitinib dilactic acid, regorafenib, cabozantinib, and crizotinib were purchased from Selleck Chemicals (Houston, TX, USA).

### 4.2. Growth Inhibition Assays

We measured the viabilities of GC cells treated with LOXO-101, entrectinib, dovitinib, dovitinib lactate, dovitinib dilactic acid, regorafenib, cabozantinib, or crizotinib concentrations of 10 µM for 48 h. On the day of the proliferation assay, the growth medium was removed, and 100 µL of fresh medium was added to each well of a 96-well plate; 10 µL of MTS solution was then added to each well. The plates were incubated at 37 °C for 2–4 h in a humidified environment with 5% CO_2_. The absorbance was measured at 490 nm using a microplate reader (Synergy 2 Multi-Mode Microplate Reader; BioTek, Winooski, VT, USA).

### 4.3. Apoptosis Analysis

NCI-N87, SNU16, MKN7, MKN28, and AGS cells were seeded into 6-well plates at a density of 1 × 104 cells/mL, then treated with 10 µM of either LOXO-101, entrectinib, dovitinib, dovitinib lactate, dovitinib dilactic acid, regorafenib, cabozantinib, or crizotinib. Cell death was assessed using an annexin V-APC/PI apoptosis detection kit (Thermo Fisher Scientific, Waltham, MA, USA) with a CytoFLEX flow cytometer (Beckman Coulter, Brea, CA, USA). The percentages of intact and apoptotic cells were calculated using CytExpert software (version 2.0; Beckman Coulter, INC., Brea, CA, USA).

### 4.4. Quantitative Real-Time PCR (qPCR) Analysis

To quantify mRNA expression, total RNA from each sample (entrectinib-treated or non-treated NCI-N87, AGS, and MKN28 cells) was reverse transcribed into cDNA using a High Capacity cDNA Reverse Transcription Kit (Applied Biosystems, Foster City, CA, USA). qPCR was performed using Power SYBR Green PCR Master Mix and a LightCycler 96 instrument (Roche Applied Science, Indianapolis, IN, USA). Transcript levels of glyceraldehyde-3-phosphate dehydrogenase (GAPDH) were used for sample normalization. The primer sequences used were as follows: NTRK1 (forward: 5′-AAA CCA GTG GAT CTG CCA AC-3′; reverse: 5′-ACG TAG CCG AAG AAA CCT CA-3′), NTRK2 (forward: 5′-TGG TGC ATT CCA TTC ACT GT-3′; reverse: 5′-CGT GGT ACT CCG TGT GAT TG-3′), NTRK3 (forward: 5′-CTC TCC CAA ATG CTC CAC AT-3′; reverse: 5′-CTA GCA GAT TCG CTC CAA CC-3′), TGFBR1 (forward: 5′-TGG GCT CTG CTT TGT CTC TG-3′; reverse: 5′-ACA AAC GGC CTA TCT CGA GG-3′), VEGFR2 (forward: 5′-GCC AAT GGA GGG GAA CTG AA-3′; reverse: 5′-TAC CTA GCT TCA GCC GGT CT-3′), CD274 (forward: 5′-TTG CTG AAC GCC CCA TAC AA-3′; reverse: 5′-TTG TCC AGA TGA CTT CGG CC-3′), ECAD (forward: 5′-TGG GCC AGG AAA TCA CAT CC-3′; reverse: 5′-GGC ACC AGT GTC CGG ATT AA-3′), CD44 (forward: 5′-AGC ATC GGA TTT GAG ACC TG-3′; reverse: 5′-GTT GTT TGC TGC ACA GAT GG-3′), CD44s (forward: 5′-AAA GGA GCA GCA CTT CAG GA-3′; reverse: 5′-TGT GTC TTG GTC TCT GGT AGC-3′), CD44v9 (forward: 5′-ACC ATC CAA CAA CTT CTA CTC TGA CA-3′; reverse: 5′-CCT TCA GAA TGA TTT GGG TCT CTT-3′), MUC6 (forward: 5′-GCC TGC AAC TAC GAG GAG AC-3′; reverse: 5′-GAT GGT GCA GTT GTC CAC AC-3′), NQO1 (forward: 5′-GCA CTG ATC GTA CTG GCT CA-3′; reverse: 5′-CAT GGC ATA GAG GTC CGA CT-3′), CTNNB1 (forward: 5′-TCA TGC GTT CTC CTC AGA TG-3′; reverse: 5′-CTC ACG ATG ATG GGA AAG GT-3′), iNOS (forward: 5′-ATG GGA GAA GGG GAT GAG CT-3′; reverse: 5′-GTC CCA GGT CAC ATT GGA GG-3′), and GAPDH (forward: 5′-TTC ACC ACC ATG GAG AAG GC-3′; reverse: 5′-GGC ATG GAC TGT GGT CAT GA-3′).

### 4.5. Immunoblot Analysis

Immunoblot analysis was conducted using standard procedures. The following commercially available primary antibodies were used: anti-panTRK (1:1000; sc7268; Santa Cruz Biotechnology, Santa Cruz, CA, USA), anti-phospho-NFkB p65 (1:1000; sc372; Santa Cruz Biotechnology), anti-ECAD (1:1000; #3195; Cell Signaling Technology), anti-β-catenin (1:1000; 610153; BD Biosciences), anti-phospho-AKT (1:500; #4060; Cell Signaling Technology), anti-AKT (1:1000; #9272; Cell Signaling Technology), anti-phospho-ERK (1:500; #9101; Cell Signaling Technology), anti-ERK (1:1000; sc514302; Santa Cruz Biotechnology), anti-NQO1 (1:1000; sc25591; Santa Cruz Biotechnology), anti-SNAIL (1:1000; #3879; Cell Signaling Technology), anti-GSK3β (1:1000; sc7291; Santa Cruz Biotechnology), anti-VEGFR2 (1:1000; #9698; Cell Signaling Technology), anti-VEGF (1:1000; sc7269; Santa Cruz Biotechnology), anti-PD-L1 (1:1000; #13684; Cell Signaling Technology), anti-TGFβRI (1:1000; sc518018; Santa Cruz Biotechnology), and anti-GAPDH (1:4000; sc32233; Santa Cruz Biotechnology).

### 4.6. Cell Migration Analysis

When cells reached confluence, a p200 pipette tip was used to scrape a straight line through the cell monolayer. The cells were then washed with phosphate-buffered saline (PBS) and further cultured with or without tepotinib in RPMI 1640. After incubation for 2–14 days, the gap width of the scratch was photographed and compared with the initial gap size at 0 days.

### 4.7. Statistical Analysis

All data were statistically analyzed using GraphPad Prism 5 software (GraphPad, La Jolla, CA, USA). All values are presented as means ± standard deviations. Differences among 3 or more groups were compared by analysis of variance, followed by the Newman–Keuls test for multiple comparisons; *p* < 0.05 was considered statistically significant.

## 5. Conclusions

The results of this study indicated that entrectinib is an attractive candidate for suppressing multiple target genes and thereby modulating VEGFR2, β-catenin, ERK, AKT, and NFκB signaling and EMT in GC patients with amplification of NTRK1-3 and VEGFR2. NTRK amplification and VEGFR2 may be useful to consider in clinical trials for entrectinib.

## Figures and Tables

**Figure 1 ijms-23-00395-f001:**
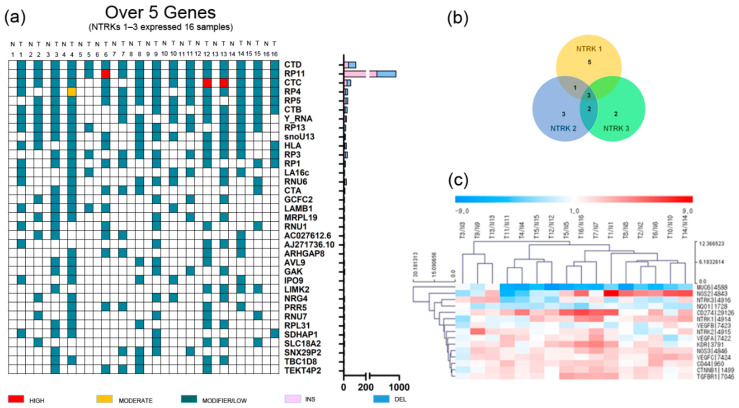
Gene alteration landscape of gastric cancer patients with amplification of neurotrophic tropomyosin receptor kinases. (**a**) Somatic indel of gastric cancer patients with amplification of neurotrophic tropomyosin receptor kinases (top 35 genes). T, tumor tissue; N, normal tissue. (**b**) Venn diagram of gastric cancer patients with amplification of neurotrophic tropomyosin receptor kinases, and (**c**) heatmap of gastric cancer patients with amplification of neurotrophic tropomyosin receptor kinases.

**Figure 2 ijms-23-00395-f002:**
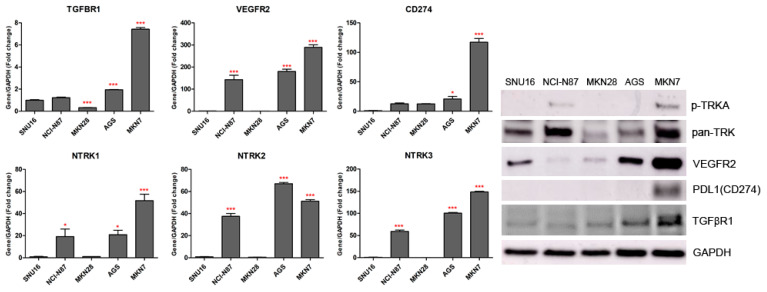
TGFBR1, VEGFR2, CD274, and NTRK1/2/3 gene and protein expression levels measured by quantitative real-time PCR and western blotting in gastric cancer cell lines. * *p* < 0.05 and *** *p* < 0.001 compared with SNU16 cell line.

**Figure 3 ijms-23-00395-f003:**
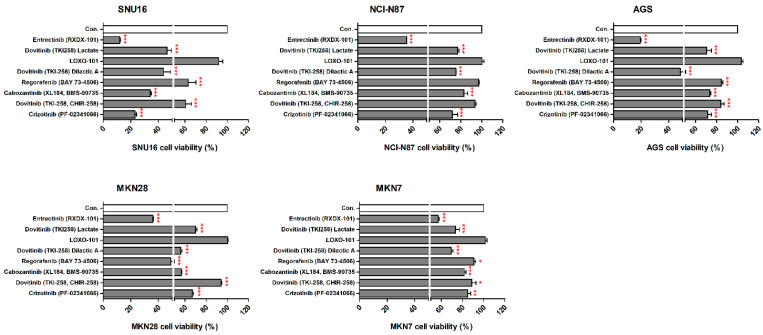
Effects of neurotrophic receptor tyrosine kinase (NTRK) inhibitors on gastric cancer cells with or without NTRK gene and protein expression. SNU16, NCI-N87, AGS, MKN28, and MKN7 cells were treated with 10 μM concentrations of NTRK inhibitors for 48 h. * *p* < 0.05. ** *p* < 0.01 and *** *p* < 0.001 compared with control group.

**Figure 4 ijms-23-00395-f004:**
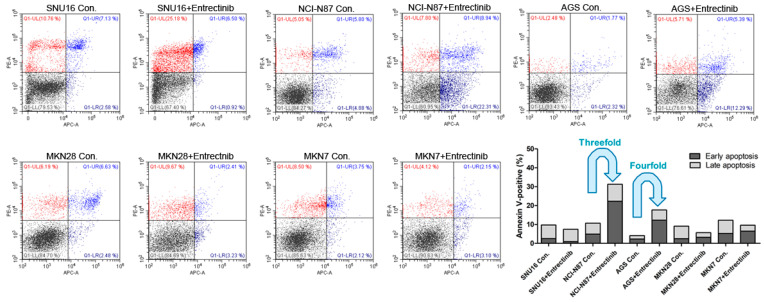
Effects of neurotrophic receptor tyrosine kinase (NTRK) inhibitors on gastric cancer cells with or without NTRK gene expression. SNU16, NCI-N87, AGS, MKN28, and MKN7 cells were treated with 10 μM concentrations of NTRK inhibitors for 48 h.

**Figure 5 ijms-23-00395-f005:**
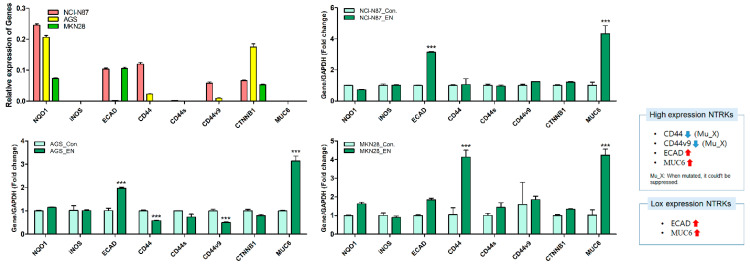
Effects of entrectinib on mRNA expression levels of EMT-related genes and antioxidant enzyme genes in gastric cancer cell lines (NCI-N87, AGS, and MKN28 cells). NQO1, iNOS, ECAD, CD44, CD44s, CD44v9, CTNNB1, and MUC6 mRNA expression levels were determined by quantitative reverse transcription-polymerase chain reaction (qPCR). *** *p* < 0.001 compared with control group. Mu, NTRK3 mutated cell lines; X, no inhibition.

**Figure 6 ijms-23-00395-f006:**
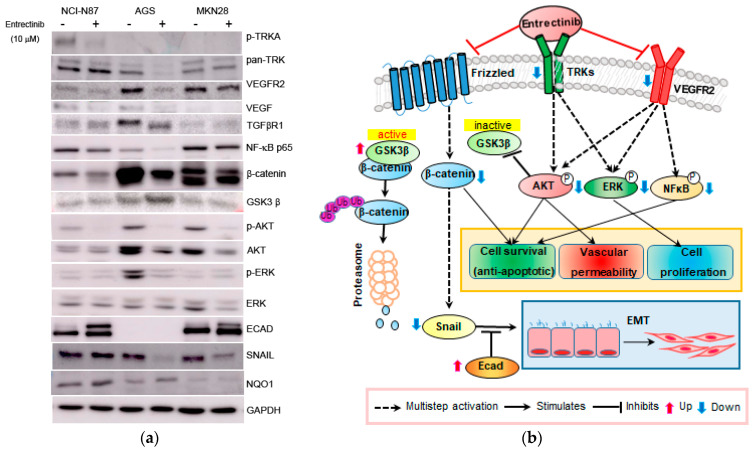
Effects of entrectinib on protein expression in gastric cancer cell lines. (**a**) Protein expression of NCI-N87 (high expression of NTRK1-3 but mutation of NTRK3), AGS (middle expression of NTRK1-3), and MKN28 (low expression of NTRK1-3) gastric cancer cell lines. (**b**) Graphic overview of entrectinib-targeted interventions in GC cancer cells. Protein expression levels of TRK, p-NFkB p65, ECAD, β-catenin, p-AKT, AKT, p-ERK, ERK, and NQO1 were determined using immunoblot analysis.

**Figure 7 ijms-23-00395-f007:**
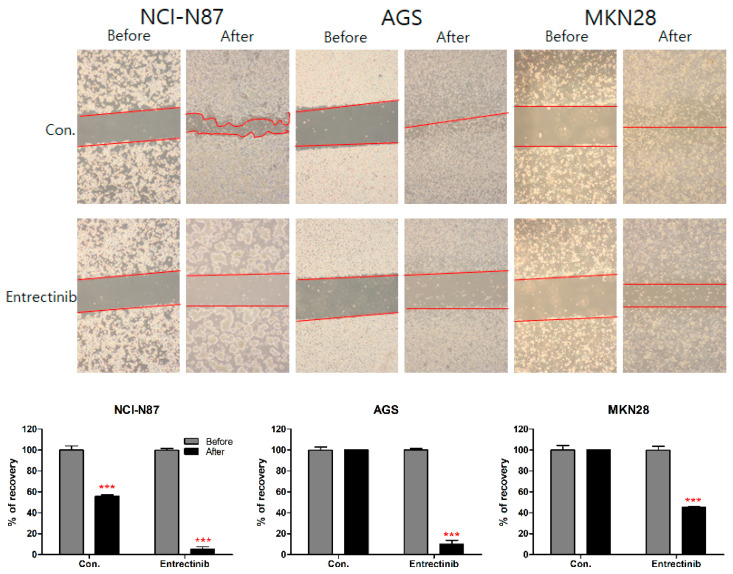
Effects of entrectinib on migration in gastric cancer cell lines. A wound-healing assay was used to assess the effect of entrectinib on the migration abilities of NCI-N87, AGS, and MKN28 cells. Entrectinib-treated gastric cancer cells showed suppressed migration ability. Con, non-treated control cells. *** *p* < 0.001 compared with before group.

**Table 1 ijms-23-00395-t001:** The characteristics of patients with NTRK overexpression (*n* = 16).

#	NTRK Expression	Sex	Age	Stage	Lauren	Reccur
1	NTRK2	M	78	IIIA	diffuse	N
2	NTRK1	F	76	IIB	mixed	N
3	NTRK3	M	53	IIIB	mixed	N
4	NTRK2	F	79	IIA	mixed	N
5	NTRKs 1–3	F	82	IIIB	diffuse	Y
6	NTRK1	F	76	IIIB	mixed	N
7	NTRKs 1–3	F	45	IIIC	diffuse	Y
8	NTRK3	M	80	IIIB	diffuse	N
9	NTRKs 2, 3	M	68	IIIA	diffuse	Y
10	NTRK1	M	74	IIB	diffuse	N
11	NTRK2	M	64	IV	intestinal	N
12	NTRK1	M	57	IIIA	intestinal	N
13	NTRKs 2, 3	F	74	IIIC	mixed	N
14	NTRKs 1–3	M	44	IIIC	diffuse	N
15	NTRK1	M	54	IIIC	diffuse	Y
16	NTRKs 1, 2	F	59	IIA	diffuse	N

Overexpression: genes differentially expressed in entrectinib-treated cells compared to non-treated cells, ≥2.0-fold change.

## Data Availability

All data presented in this study are available in the main body of the manuscript.

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
