# Peer review of "Entrectinib Induces Apoptosis and Inhibits the Epithelial–Mesenchymal Transition in Gastric Cancer with NTRK Overexpression"

_ijms, 2021, doi:10.3390/ijms23010395_

Round 1

Reviewer 1 Report

By screening the mRNA sequencing data on gastric cancer patients, the authors found the up-regulation of NTRK in gastric cancers and identified the downstream molecules related to the up-regulation. After test of several NTRK inhibitors, the authors found the different response to these inhibitors among cell lines and determined entrectinib treatment are more powerful. Then the authors studied the cell apoptosis and migration, as well as the related genes. Overall, NTRK expression and targeting has been studied in the present study. It’s a good model as shown here to begin a study from patients data. However, there are several issues needed to be addressed.

Major:

1) The authors found that 47% patients with GC harbor the up-regulation of NTRK1-3, which is much higher than reported. Do the other studies use RNA-seq to analysis the expression of NTRK or other methods, such as immunohistochemistry (IHC)? As mRNA is still needed to be transfected and post-transcription regulation may change the protein expression of the genes? The authors should verify their findings using IHC staining? Or they should narrow their conclusion.

2) As introduced by the authors, KM12SM is a positive control, we can find the up-regulation of NTRK1 and NTRK2 in these cells are not obvious. However, NTRK3 is much higher up-regulated. The authors may need to identify which receptor is the target of these inhibitors.

3) Entrectinib significantly inhibit the SNU16 cell growth but does not affect the cell apoptosis. The authors may need to explain the underlying mechanisms.

4) The authors concluded that CD44 is responsible for the resistance to Entrectinib in GC cells. However, the authors did not perform mechanical experiments to prove that CD44 indeed mediate the drug resistance.

5) In all figures, the authors did not indicate the statistical analysis and the p value.

Minor:

1) There are some typos in the manuscript needed to be carefully checked.

2) As the authors focused on the NTRK receptors, protein expression is needed to verify.

Author Response

Major:

1) The authors found that 47% patients with GC harbor the up-regulation of NTRK1-3, which is much higher than reported. Do the other studies use RNA-seq to analysis the expression of NTRK or other methods, such as immunohistochemistry (IHC)? As mRNA is still needed to be transfected and post-transcription regulation may change the protein expression of the genes? The authors should verify their findings using IHC staining? Or they should narrow their conclusion.

Answer: Since NTRK detection is low in DNA-seq and IHC, there is a trend to use more sensitive RNA-seq in clinical trials. The study of NTRK1-3 mainly looked at fusion by RNA-seq, but we saw overexpression of NTRK1-3.

2) As introduced by the authors, KM12SM is a positive control, we can find the up-regulation of NTRK1 and NTRK2 in these cells are not obvious. However, NTRK3 is much higher up-regulated. The authors may need to identify which receptor is the target of these inhibitors.

Answer: Thank you for the appropriate comments. We changed the figure 2 because it could be confusing. Entrectinib (RXDX-101, NMS-E628) is an orally bioavailable pan-TrkA/B/C inhibitor.

We deleted the words “KM12SM (NTRK positive control; human colon cancer cell line, possess tropomyosin 3–neurotrophic receptor tyrosine kinase 1 (TPM3–NTRK1) fusion gene) in the figure 2 legend.

3) Entrectinib significantly inhibit the SNU16 cell growth but does not affect the cell apoptosis. The authors may need to explain the underlying mechanisms.

Answer: According to the referee’s comment, we conducted western blot analysis in SNU16. β-catenin has been also implicated in cell growth control and apoptosis (Ahmed et al., 1998; He et al., 1998; Orford et al.,1999; Tetsu and McCormick, 1999; Zhu and Watt, 1999), it might be expected to play a role in regulating growth by E-cadherin. E-cadherin inhibits β-catenin/TCF signaling and SW480 tumor cell growth without noticeable changes in cytosolic/nuclear levels of β-catenin protein, the major mechanism thought to control β-catenin signaling (Polakis, 1999).

4) The authors concluded that CD44 is responsible for the resistance to Entrectinib in GC cells. However, the authors did not perform mechanical experiments to prove that CD44 indeed mediate the drug resistance.

Answer: We appreciate the additional comments and we deleted confusing paragraph. We deleted the sentences “Notably, CD44, CD44s, and CD44v9 gene levels were only decreased in entrectinib-treated AGS 25 cells, while these genes were upregulated in entrectinib-treated MKN28 cells. Together, these results suggest that entrectinib resistance is closely associated with CD44 during apoptosis induction in gastric cancer cells.” in abstract.

5) In all figures, the authors did not indicate the statistical analysis and the p value.

Answer: According to the referee’s comment, we added the statistical analysis and the p value in figures and figure legends.

Minor:

1) There are some typos in the manuscript needed to be carefully checked.

Answer: According to the referee’s comment, we checked in the manuscript.

2) As the authors focused on the NTRK receptors, protein expression is needed to verify.

Answer: We identified pan-NTRK and p-NTRKA as WB in the paper.

Reviewer 2 Report

I think the Authors in this study would like to describe the effect of some drugs on gastric cancer samples.
In reality, it is not clear what kind of samples were used (cell cultures? 16 surgical samples? 34 surgical samples?) and there is no control to validate the results (healthy tissues or non-mutated cell lines). Furthermore, the results are not supported by statistical analysis (the k and / or p value is not reported) and there is no order in describing the various molecules analyzed.
Finally, the work is not properly ordered and structured, resulting in confusion and disorientation.

For more details, please find attached file.

Author Response

Answer: We wrote an answer in the attached file.

Round 2

Reviewer 1 Report

The manuscript has been improved according to the comments.

Reviewer 2 Report

The paper has improved.

This manuscript is a resubmission of an earlier submission. The following is a list of the peer review reports and author responses from that submission.

Round 1

Reviewer 1 Report

This manuscript by Sohn et al. suggests an effect of the Trk-inhibitor Entrectinib on epithelial-mesenchymal transition (EMT) and WNT/beta-catenin signaling in gastric cancer (GC) tumor cells.

The study originates from the sequencing of 16 GC patients and it can be of potential interest especially due to the high incidence of GC in Korea. However, the rest of the results are merely observational. Overall the authors try to make sense of different results in different cell lines without convincing evidence. Ultimately the study reads like a list of observations relative to molecules involved in cancerogenesis without addressing any specific mechanism.

Specific comments.

1-In figure 1, statistics and more details about the elaboration of results are needed.

2- Results in figure 2 would be more convincing if protein levels were shown and not just RNA, which may or may not reflect real protein expression. This is even more important in light of the results shown in figure 6 where only three lines are shown.

3- The paper would be significantly stronger with in vivo animal-based work.

4- In figure 6, a positive control for Trk phosphorylation is missing.

5- Discussion: the second part reads like an introduction on beta-catenin signaling. Results are not critically discussed. The introduction and discussion completely miss a part in which the role of different Trks is illustrated. 

Author Response

Reviewer #1:

1-In figure 1, statistics and more details about the elaboration of results are needed.

Answer: We agree with reviewer’s concern. We changed the sentences in the line 70-79, page 2. “We performed RNA sequencing analysis of 34 GC patients, then selected 16 GC patients with amplification of NTRK1-3 (defined as ≥ 2.0 fold change) [13]. In GC patients with NTRK amplification, expression levels of NOS2, CD274 (also known as B7-H1 and PD-L1), KDR (VEGFR2), CD44, VEGFA–C, CTNNB1, and TGFBR1 were upregulated, while expression levels of MUC6 and NQO1 were downregulated (Figure 1). These patients did not exhibit fusion or mutations of NTRK1-3.” à “We performed RNA sequencing analysis of 34 GC patients (mostly tumor stage III~IV), then selected 16 GC patients with amplification of NTRK1-3 (47%), defined as ≥ 2.0 fold change (Figure 1a) [13]. These patients did not exhibit fusion or mutations of NTRK1-3. The Venn diagram was prepared to identify the upregulated NTRK1-3 genes in 16 GC patients. Nine patients were upregulated NTRK 1, NTRK2 in 9 patients and NTRK3 in 7 patients (Figure 3b). Three patients were found to upregulate all three NTRKs. In 16 GC patients with NTRK amplification, expression levels of NOS2, CD274 (also known as B7-H1 and PD-L1), KDR (VEGFR2), CD44, VEGFA–C, CTNNB1, and TGFBR1 were upregulated, while expression levels of MUC6 and NQO1 were down-regulated (Figure 1c).” has been added.

We changed the Figure 1.

2- Results in figure 2 would be more convincing if protein levels were shown and not just RNA, which may or may not reflect real protein expression. This is even more important in light of the results shown in figure 6 where only three lines are shown.

Answer: Sorry. TGFBR1, KDR (VEGFR2), CD274 ab are not stocked and must be purchased. It takes 3-4 weeks for us to order and receive. We did check TRK. SNU16 did not express NTRK 1-3 gene, but TRK protein was expressed when pan-TRK ab was used. In MKN7, NTRK 1-3 expression was high, but TRK protein was not expressed. Therefore, two cell lines were excluded from subsequent experiments because it was difficult to explain.

We added the sentences in the line 98-102, page 3. “SNU16 did not express NTRK 1-3 gene, but TRK protein was expressed when pan-TRK ab was used. In MKN7, NTRK 1-3 expression was high, but TRK protein was not expressed (Figure 2 and 3). Notably, TRK protein was suppressed in entrectinib-treated SNU16 and AGS cells, while this protein was not suppressed in entrectinib-treated NCI-N87 and MKN28 cells.”

We changed the Figure 3.

3- The paper would be significantly stronger with in vivo animal-based work.

Answer: Thank you for your kind review. We all agree on your opinion. We also want to perform in vivo animal-based work, but it takes a lot of time and money to prepare. Currently, we are analyzing the gastric cancer panel of about 300 gastric cancer patients, so it is difficult to proceed directly with animal experiments. sorry.

4- In figure 6, a positive control for Trk phosphorylation is missing.

Answer: We agree with reviewer’s concern. Expression of p-TRKA was deleted from the figure 6 because it was absent in the cells.

Positive control [KM12SM cell line: human colon cancer cell line, possess tropomyosin 3–neurotrophic receptor tyrosine kinase 1 (TPM3–NTRK1) fusion gene]

5- Discussion: the second part reads like an introduction on beta-catenin signaling. Results are not critically discussed. The introduction and discussion completely miss a part in which the role of different Trks is illustrated.

Answer: Thank you for the appropriate comments. We added the sentences in the line 186-191, page 6. “TRKA signaling pathway is leading to cell growth and proliferation via ERK signaling [21, 22]. TRKB is activate the PI3K, PLCγ, and RAS-ERK pathway, resulting in cell differentiation and survival [21, 22]. TRKC is activate the PI3K/AKT pathway, preventing apoptosis and increasing cell survival [21, 22]. Indeed, we observed that the AGS (high expression of NTRK1-3) gastric cancer cells were more finely sensitive to the pan-TRK inhibitor entrectinib than NCI-N87 (high expression of NTRK1-3 but mutation of NTRK3).”

We hope that these revisions fulfill all the referees’ concerns. On behalf of the research team I would like to thank you for your efforts in improving our manuscript.

Best regards,

Dae Young Zang, M.D., Ph.D.

Professor

Department of Hematology-Oncology, Hallym University Medical Center, Hallym University College of Medicine

Reviewer 2 Report

This study tried to evaluate entrectinib, a TRK inhibitor, inhibits apoptosis and EMT in gastric cancer cell lines. The topic might be relatively novel but several results seems not reasonable.

  1. Were NTRK1, NTRK2, and NTRK3 also mutated or amplified in gastric cancer of Korea? The association should be confirmed to ensure that TRK inhibitors were worthy to evaluate in gastric cancer. Authors reported 16/34 (47%) gastric cancer patients revealed RNA overexpression (amplification?) of NTRK3 genes. It is much higher than those data reported in previous studies of several cancer types (~14.5-14.8%, JCO Precis Oncol. 2018; 2: PO.18.00183, Precision and Future Medicine 2017;1(3):129-137.) Was there any bias for collection of those samples?
  2. Moreover, “NTRK mutation or amplification, such as non-fusion NTRK alteration, has been associated with a lack of response to NTRK inhibitors…. (line 54)”. If those patients with overexpressed NTRK mRNA did not exhibit fusion or mutations of NTRK1-3 (line 75), were those genes acquired form RNA seq considerable for subsequent experiments of NTRK inhibitors? (Figure 1)
  3. Similarly, five gastric cancer cell lines were recruited for comparison of those NTRK inhibitors. Were those cells containing fusion of NTRK1-3? If not, most NTRK inhibitors did not reduce noticeable cell viability were reasonable. Only entrectinib resulted in lower cell viability (> IC 50) in 4 cells, including two NTRK low cells, SNU16 (most sensitive) and MKN28. It was truly hard to deduce the inhibition of cell viability to NTRK1-3 completely according those results (Figure 3).
  4. In Figure 4, entrectinib induced higher apoptosis in two cells with higher VEGFR2 expression, and it also implicated that entrectinib cytotoxic function might be caused from other signaling pathway other than NTRK1-3.
  5. In figure 5-6, those signaling molecules were complicated and not representative for specific pathways, such as EMT, more detail classification should be contributed to make those data more understandable.
  6. Several typographic and grammatical mistakes need to be corrected and revised more precisely.

Author Response

Reviewer #2:

  1. Were NTRK1, NTRK2, and NTRK3 also mutated or amplified in gastric cancer of Korea? The association should be confirmed to ensure that TRK inhibitors were worthy to evaluate in gastric cancer. Authors reported 16/34 (47%) gastric cancer patients revealed RNA overexpression (amplification?) of NTRK3 genes. It is much higher than those data reported in previous studies of several cancer types (~14.5-14.8%, JCO Precis Oncol. 2018; 2: PO.18.00183, Precision and Future Medicine 2017;1(3):129-137.) Was there any bias for collection of those samples?

Answer: We agree with reviewer’s concern.

> JCO Precis Oncol. 2018; 2: PO.18.00183 assessed 13,467 samples available from The Cancer Genome Atlas (adult tumors) and the St Jude PeCan database (pediatric tumors) for the prevalence of NTRK fusions, as well as associated genomic and transcriptomic co-aberrations in different tumor types. This paper used data from analysis of various carcinomas of different races.

> Precision and Future Medicine 2017;1(3):129-137 assessed 1,250 samples using the NGS cancer panel. NTRK amplification was detected in 28 cases of various types of cancer. NTRK protein was overexpressed in only 14.8% of these patients. This patients age is 59 (2-74) years. This paper used data from various types of cancer analysis in Koreans. The selection criterion is DNA copy number >4.

> This study, the number of gastric cancer patients we analyzed was small, and many of them had overexpressed NTRK 1-3 expression. There is a limit to the interpretation of the results because of the small number of data. Unlike other papers, normal and tumor were analyzed as a pair in the surgical tissue of 34 Korean gastric cancer patient, mostly tumor stage III~IV, Age 69 (44-87) years. The selection criterion is RNA-seq fold change > 2.

We added the sentence in the line 72-73, page 2. “There is a limit to the interpretation of the results because of the small number of data.”   

  1. Moreover, “NTRK mutation or amplification, such as non-fusion NTRK alteration, has been associated with a lack of response to NTRK inhibitors…. (line 54)”. If those patients with overexpressed NTRK mRNA did not exhibit fusion or mutations of NTRK1-3 (line 75), were those genes acquired form RNA seq considerable for subsequent experiments of NTRK inhibitors? (Figure 1)

Answer: We selected genes associated with gastric cancer from among the genes from RNA-seq analysis and used them in subsequent experiments.

  1. Similarly, five gastric cancer cell lines were recruited for comparison of those NTRK inhibitors. Were those cells containing fusion of NTRK1-3? If not, most NTRK inhibitors did not reduce noticeable cell viability were reasonable. Only entrectinib resulted in lower cell viability (> IC 50) in 4 cells, including two NTRK low cells, SNU16 (most sensitive) and MKN28. It was truly hard to deduce the inhibition of cell viability to NTRK1-3 completely according to those results (Figure 3).

Answer: We agree with reviewer’s concern. Five gastric cancer cell lines does not have fusion of NTRK1-3.

We added the WB image in Figure 3.

We added the sentences in the line 98-102, page 3. “SNU16 did not express NTRK 1-3 gene, but TRK protein was expressed when pan-TRK ab was used. In MKN7, NTRK 1-3 expression was high, but TRK protein was not expressed (Figure 2 and 3). Notably, TRK protein was suppressed in entrectinib-treated SNU16 and AGS cells, while this protein was not suppressed in entrectinib-treated NCI-N87 and MKN28 cells.”

  1. In Figure 4, entrectinib induced higher apoptosis in two cells with higher VEGFR2 expression, and it also implicated that entrectinib cytotoxic function might be caused from other signaling pathway other than NTRK1-3.

Answer: We agree with reviewer’s concern. We added the sentences in the line 126-128, page 4. “Entrectinib induced higher apoptosis in two cells with higher VEGFR2 expression, and it also implicated that entrectinib cytotoxic function might be caused from VEGFR2 signaling pathway other than NTRK1-3.”

  1. In figure 5-6, those signaling molecules were complicated and not representative for specific pathways, such as EMT, more detail classification should be contributed to make those data more understandable.

Answer: We agree with reviewer’s concern. We added the words in the line 142-143, page 4. “EMT inhibitory genes such as ECAD and MUC6 genes”

We added the pathway image in Figure 5.

We added the pathway image in Figure 6.

  1. Several typographic and grammatical mistakes need to be corrected and revised more precisely.

Answer: Thank you for the appropriate comments. Even though we have already edited the English by an English editing institution, we re-edited the English by another native English speaker following the reviewer’s comment. We hope the English quality is sufficient this time.

The English in this document has been checked by at least two professional editors, both native speakers of English. For a certificate, please see:http://www.textcheck.com/certificate/EeCuXO

We hope that these revisions fulfill all the referees’ concerns. On behalf of the research team I would like to thank you for your efforts in improving our manuscript.

Best regards,

Dae Young Zang, M.D., Ph.D.

Professor

Department of Hematology-Oncology, Hallym University Medical Center, Hallym University College of Medicine

Round 2

Reviewer 1 Report

The Authors have performed cosmetic changes to the manuscript and not addressed any substantial concern previously expressed. While I sympathize for the lack of resources to put into this work, I cannot really accept that as a justification not to do experiments that will raise the quality.